# ‘Safety First’: Residents, Families, and Healthcare Staff Experiences of COVID-19 Restrictions at an Irish Residential Care Centre

**DOI:** 10.3390/ijerph192114002

**Published:** 2022-10-27

**Authors:** Michael Connolly, Anita Duffy, Mary Ryder, Fiona Timmins

**Affiliations:** 1School of Nursing, Midwifery & Health Systems, University College Dublin, Belfield, D04 V1W8 Dublin, Ireland; 2Education & Research Centre, Our Lady’s Hospice & Care Services, Harold’s Cross, D6W RY72 Dublin, Ireland

**Keywords:** resident, residential care, family, COVID-19, infection control, restrictions, lockdown

## Abstract

The COVID-19 pandemic and the need to stem the transmission and protect the most vulnerable in society led to infection control restrictions effectively locking down an entire country. These restrictions were also imposed on residential care settings for older people, initially in March 2020, and subsequently at varying times throughout the year that followed. Furthermore, the restrictions led to the suspension in all visiting to residents expect in exceptional circumstances and on compassionate grounds. The purpose of this research study was to develop an understanding of the experience of residents, their families, and carers in an Irish residential care setting during the COVID-19 lockdown. Data were collected in a residential care centre for older people in Ireland, using semi-structured interviews of residents, family members and staff. Interviews were conducted in person for residents and virtually for family members and staff. In total 29 people were interviewed. Data were analysed using Braun and Clarke’s thematic data analysis approach. Four themes and three subthemes were developed from the data. The main themes were ‘difficult but safe’, ‘communication’, ‘staff going above and beyond’, ‘what about the future?’ Residents, families and staff of the residential care setting had to manage and cope with the challenges of the restrictions imposed during COVID-19 lockdown. This study highlighted the negative impacts of visiting restrictions on staff, residents and their family members during the COVID-19 lockdown.

## 1. Introduction

The COVID-19 pandemic restrictions on movement changed the way healthcare is provided in residential care settings (RCS) worldwide. Nursing home residents are perceived as a vulnerable group within society [1]. According to the Health Protection Surveillance Centre [2] 2481 deaths of nursing home residents have been reported in the Republic of Ireland due to COVID-19. Findings from a study on how and where people died in the Republic of Ireland reported that approximately 30,000 people die each year in Ireland, with 60% of these deaths occuring in a RCS [3]. Older people living in RCS expect to be cared for safely, in a person-centred manner with choice and freedom to exercise their autonomy [4]. These expectations were challenged during the COVID-19 enforced lockdown restrictions.

COVID-19 caused by the Coronavirus, originated in Wuhan city in China, and was first reported to the World Health Organisation (WHO) in December 2019. In January 2020, the WHO announced the outbreak as a Public Health Emergency. On March 11th, the WHO declared the coronavirus crisis as a pandemic, and by then the virus had reached Ireland. In January 2020, to curtail the virus, the Irish Government established the National Public Health Safety Team (NPHET) and on 13 March 2020, the Government advised by NPHET directed the Nation to self-isolate and socially distance from others. Older citizens and those with underlying health concerns were told to cocoon in their homes and nursing home providers were instructed to shield residents by restricting physical visits to continue to provide a safe service.

In the absence of a treatment or vaccine, containment and mitigation measures were deemed crucial public health interventions to slow down the spread of the virus and reduce the impact of COVID-19. Ireland entered a stringent three-month national lockdown.

Infection prevention and control (IPC) restrictions needed to ensure residents’ safety can lead to social isolation and loneliness [5], with significant negative physical and psychosocial impacts on the lives of those in RCS. Negative effects of social isolation include physical deterioration, anxiety, adjustment disorders, depression, stress and insomnia [6]. Sadly the early part of the COVI|D-19 pandemic saw an increase of inappropriate ageist comments trending on social media platforms suggesting older people were a burden on healthcare systems and society with the use of terms such as ‘#boomerremover,’ [7]. Media discussions debated the ethics of medical resource allocation for older people, implying older people are less valuable than younger people. Ageist discriminatory remarks about older people and COVID-19 contributed to a sense of worthlessness with a negative attitude towards older people being a burden on the state, adding little value to society.

An initial review of the literature in the early days of the pandemic indicated a dearth of research on the impact of COVID-19 on residents, their families and healthcare staff; however, there were a plethora of editorials and commentaries available. A report from The Irish Longitudinal Study on Ageing (TILDA) led by Romero-Ortuno stated “Nursing home residents have been disproportionately affected by the COVID-19 pandemic” [8] and welcomed research including this vulnerable group. Prior to commencing this study, the researchers additionally found no study exploring the impact of COVID-19 restrictions on families of residents with limited published research on staffs’ experiences of this phenomenon.

The fundamental aim of our research study was to understand the experiences of residents, their families, and carers in a residential care setting during the COVID-19 lockdown.

## 2. Materials and Methods

### 2.1. Design

A descriptive qualitative research design [9] was chosen as an appropriate methodology to develop an understanding of the participants’ experiences of the phenomenon. The researchers used individual semi-structured interviews to explore residents, family members and staff’s experiences of the lockdown restrictions imposed during COVID-19 pandemic.

### 2.2. Sample & Recruitment

The target population included individuals who lived in a residential care facility, family members of the residents and staff who worked in the facility. A convenience sample was generated in the facility, with information provided to residents, family members and staff in written form. Inclusion and exclusion criteria were identified for each participant group (Table 1). Demographic details of the individuals interviewed are set out in Table 2, Table 3 and Table 4 respectively.

### 2.3. Data Collection

Following ethical approval from the University Hospital Ethics Committee (RS21-039) data collection commenced. Data were collected through individual semi-structured interviews. Each interview was digitally recorded and transcribed verbatim. Interviews with residents were conducted face to face, with social distancing maintained and both interviewer and participant wore face masks. Interviews with relatives and staff were conducted virtually. Interviews lasted between twenty to forty minutes.

### 2.4. Data Analysis

Data were analysed using the six-step method developed by Braun and Clarke [10,11,12]. This initial reading was used to develop a coding scheme that was applied to all interview transcripts. Open codes were allocated to each of the interviews reflecting the substance of each interview. The interview transcripts were coded and compared with each other within the groups, looking for similarities from the data. Similar codes were grouped together to form categories. Categories were then compared with other categories and reduced to form initial themes. After initial themes and findings were developed, these were finalised through a final read of the full data set. All steps were conducted as an iterative and reflective collaboration between two of the research team researchers, who met regularly to discuss the data analysis process and findings. NVivo was used to manage all interview data. Following the analysis process, four major themes with a number of sub-themes were developed.

## 3. Results

In total, 29 individuals were interviewed representing residents (*n* = 9), family members of residents (*n* = 12) and staff (*n* = 8). Of the residents, the mean age was 72 years. Three male and six female residents were interviewed. The mean age of the relatives interviewed was 55 years, and included eight male and five female relatives. Six family members had their mother in the RCS, and one son also had an aunt, four family members had a spouse, and two had a sister living in the RCS. The mean age of the staff interviewed was 49 years, with the majority having more than 10 years’ experience working in care of older person services. The level of education of staff ranged from Level 5 to Level 9 (NQF).

The experiences of residents, relative and staff of COVID-19 restrictions can be summarized into four themes and three subthemes (Table 5). Implicitly, the themes were reflective of the challenges and responsibilities of living with the visitor restrictions applied during different periods of the COVID-19 pandemic.

### 3.1. ‘Safety-First’

Each of the participants highlighted the difficulties they experienced when COVID-19 restrictions were introduced. Among residents (R) they accepted the reality of the lockdown restrictions to ensure their safety. All the residents in the study said they felt safe.

‘Definitely felt safe. Yeah, you were safe and they were safe. At first it was hard ‘cos of the masks and stuff, but you just got used to it.’(R1)

‘But… I felt safe in here during Covid … the way they look after you, if they got wind of a sniff they would isolate you, nothing could get you, a germ wouldn’t be allowed in here. So, I felt very safe here over the Covid.”(R8)

For some family members of residents (F), this sense of safety was central to their concerns. For them, it was simply about putting their family member’s safety first:

‘You have to put their safety first and you have to do this … to care for our relatives and the staff you know. I think everything was done very well, and as I said, they’re both still healthy and safe, so that’s … that’s the main thing. You know. It’s, its short-term, short-term inconvenience, really.’(F4)

While for others safety was imported; but, also the comfort that this instilled in them was reflected during the interviews:

‘So yeah no, I felt very comforted. I felt she was extremely safe. I felt she was probably in the safest place in the country. And she was safer than most citizens in the country. That was my personal view. So, I was comforted by that.’(F10)

While staff (S) also felt a strong desire to protect the residents from exposure and risk:

‘…but those that did know were so grateful for what we were trying to do in trying to protect them, and they were as grateful and understanding you know…Yeah, we need to just do what we need to do and leave the room, leave your space and they, were…they kept you going in that sense. So, there’s the residents understanding, and colleagues understanding, and just that kind of thing, we were all feeling in the same boat. You know, in that way yeah. Yeah.’(S2)

### 3.2. Communication

The need for clear communication of guidance towards visit restrictions during COVID-19 was highlighted by all participants, although most particularly by family members and staff, arguing communication could have been better and particularly stressing visiting policy changes were slow to be transmitted by the RCS.

‘Families were the last to know anything, they were at the bottom of the pyramid, and sometimes you’d come in and the visiting teams would hand you a letter saying visiting times may change or be reduced, and you’d get such a shock with this letter, given out on a Friday when there was no leeway for families, families would have nowhere to go…’(F2)

‘For me there was a lot of miscommunications, and I know everyone was working on the back of something new every day and I imagine that’s why, one department might be telling another department this is happening and one department telling another department this visitor is coming and that wouldn’t have been communicated and there was a couple of times some of us were escorted off the premises, even though we had had the conversation with a number of staff.’(F11)

‘Communication … Yeah, it was it … there was a break in the chain of communication at times. As well as that, the policy changes and the communication about those policy changes was slow.’(S8)

#### 3.2.1. ‘The Window Visits’

Visits for residents were facilitated initially with window visits, where their family members and friends could communicate with and see the residents through a window, with family or friends positioned outside. For many residents these visits were less than satisfactory due to problems with their hearing and feeling cold when the window was open for the visit. Family members were also critical of window visits.

‘I mean the hardest of all I think was the window visits … they were upsetting and frustrating for her and heart-breaking for us … and some days she didn’t even want to talk to you but she just wants to know that you’re there, you know, so you have a hunch.’(F1)

‘And then the most difficult part was standing at the window in the cold looking in and trying to get the attention of the staff. That was probably the most horrific part of my whole journey with this thing. It was very rarely I didn’t go home crying.’(F2)

‘But they come to see you. Yeah. And you would have to move your chair and you were feeling cold yourself at that time… with the windows open.’(R2)

‘Initially the idea of the window visit was fantastic, I mean you’d take anything if you’ve got nothing. But, I sat at that window, I’m never cold; but, I perished. I wouldn’t pretend to my family and friends, because it was so good to see them.’(R8)

Indeed, some staff also felt that window visit were not the most optimal way for families to connect, with some finding this type of interaction difficult:

‘…when they have the visits on at the window. That was very tough I found that very tough.’(S3)

#### 3.2.2. Using Technology Helped

The use of technology provided an innovative way to enhance communication between residents and their family members. Many of the residents living in the care facility did not themselves own or use a mobile telephone or other handheld communication device. So, at first staff would use their own mobile phones to enable telephone or video conversations between the residents and their families and later organisational handheld devices such as iPads supported this activity. Most family members were grateful for the opportunity to use technology specifically to enhance window visits when their relatives complained of the cold, or difficulties hearing them.

‘We’d FaceTime, we’d all just chat for while on FaceTime. So that was good. So that was useful. That was a big plus.’(F5)

‘There was a lot of telephone conversation … a lot of using the mobile and it took a while before she could get used to it.’(F7)

‘She has, she has an iPad. So, it’s okay for me when I get her to use the WhatsApp, the WhatsApp group to communicate … and the staff are there to help her.’(F8)

However, some of the residents were not interested in the new communication technology

‘Technology is beyond me.’(R5)

‘I went to night school years ago to learn how to use computers, but I couldn’t take it in, and the kids just kept getting annoyed, I kept forgetting what to do, I just gave it up.’(R4)

Although for other residents the use of technology to communicate with the outside world was a ‘life saver’ helping them feel connected to their family. This way of communicating was also made easier with the assistance of staff who went above and beyond their duty to sit with residents and demonstrate how the technology worked:

‘I’m illiterate as far as computers are concerned; but, she sat down with me and showed me how to use it, it was great company.’(R6)

One of the residents’ sons found the use of technology negatively affected him emotionally

‘…the toughest thing for me was the video calls, seeing my mom’s distress, every day, every single day.’(F12)

Although his sister agreed, she reminded him the video calls were not as difficult as the window visits were for their family,

‘I gave birth to my first and only child and the restrictions suddenly came in and suddenly no-one was allowed in and in March I was allowed to bring my daughter outside my mums room and she wasn’t allowed to touch her or hold her or anything, and that was extremely difficult. For me it hammered home, is this how it was going to be, she was a grandmother and couldn’t hold her granddaughter, I couldn’t hold her and she couldn’t hold me and it was very very distressing’.(F11)

He then reflected,

‘…it was really well done, mum had a dedicated iPad, and they gave her plenty of time on it… each day. It was a life saver I guess, because that’s all we got. It turned out to be essential.’(F12)

For staff the use of technology brought mixed feelings with some seeing the value of using technology to enable contact between residents and their families, while others felt that the lack of sufficient equipment meant that some residents used the available iPads for long periods at time and needed lots of help with this activity:

‘Oh, the iPad was very helpful.’(S4)

‘…like we only had two devices and iPads … with 23 residents. And they would hog the iPads like an hour, an hour and a half. And one staff member would have to help…’(S7)

#### 3.2.3. Feeling Disconnected

Residents and staff felt the COVID-19 restrictions resulted with a sense of disconnection from reality, from their families and from society in general. Residents reported feeling disconnected from other individuals who lived in the facility whom they would have had regular contact with on the corridors, at mealtimes or during planned activities before the COVID-19 restrictions:

‘I think it affected everyone in this place very deeply. I would be aware of people in here who couldn’t understand, as far as they were concerned, I imagine, they would think they were abandoned because their people couldn’t come in, they saw nobody. They are going to think they have been dumped and we were isolated in our rooms, so they don’t get to see anyone. People in hospital are very vulnerable.’(R8)

For staff though, the sense of disconnect was from the wider facility community as the residential space for older people was completely isolated from other service areas, with staff required to remain in their work environment throughout their shift, and to take all meal breaks in the unit. Consequently, staff claimed to be disillusioned by the disconnection from their peers and senior managers:

‘And just for management to actually realize, like, what your staff are doing, like, you know, it’s easy to say “Oh, you are great” we’re not asking for, you know, something big, just like, just when you call in, on or like, listen to us. I’m not in a good place, or, you know, this has happened… or to just actually speak to the person, like a human…’(S1)

‘Now, there was just one manager, one site nurse at night that comes in to do the rounds, mask on, and yea, social business, like how are you? And that was amazing. Yeah, just one, the other ones never left their office. We didn’t mind, but we thought it was a lot of the act of you know… its kindness to say, how are you and have just some human contact.’(S8)

For families there was a sense both staff and senior management needed to demonstrate more humanity and empathy when communicating with them and their family members and indeed senior management needed to be empathetic towards their staff also.

‘I think senior management could have been a bit more vocal I think for what was happening sometimes is senior management were very formal lacking empathy, we were the family and while we were disconnected we found the staff on the shop floor could have done with more support.’(F6)

One family member claimed to be most upset by the fact she was detached from her husband during the pandemic, and she maintained connecting with him depended on the staff who were on duty each day:

‘I did feel the so-called managers didn’t understand how horrific it was for the residents, for the families emotionally…I felt there was a bit of a detachment you know, it was like you’re not allowed in, you’re not seeing your husband.’(F2)

### 3.3. Staff Going above and Beyond

There was no doubt from residents and their family’s staff were fully committed to their job. This commitment was shown in the way residents and their families spoke about the levels of support they experienced and the sacrifices they knew staff experienced when doing their utmost to care for and protect the residents in the RCS.

‘…the staff here could not do enough for anyone, I said it to my friend … I said they are working very hard here and they don’t have enough of people to help, they need more helpers and she said they are working very hard…’(R4)

‘I thought to myself, I have to get out of here, the idea of being locked in here… but then I finally settled… I’m convinced this is the best place for me to be now. I know I’m well cared for in here, I know how much they care about me. I’d be vain if I said they love me; but I know some of them do love me. This place will do me grand!’(R8)

‘And I just got to know, the wonderful thing about it all was I got to know the staff so well, and who were the real heroes, who had high levels of empathy and who had high levels of compassion and who had low levels and, and, to be fair, he has come out of this through all of this extraordinarily well, you know.’(F2)

‘And the medical expertise, the nursing care … that went above and beyond we felt they really tried their best in really difficult times. So, we really could not stress enough that the medical team are great. The nursing team really looked after our mother and the doctors looked after our mother throughout COVID and continually do, the health care teams’(F6)

### 3.4. What about the Future?

Staff and family members highlighted the need for better communication in the event of future infection prevention and control restrictions in RCS. Suggestions focussed on how better to use technology to support interactions between residents and their families.

‘… was thinking, if there were, you know, yeah, definitely video call facilities and rooms, you know, just that they would have been, like, agreed or set times for in person visit in a separate room.’(F6)

‘I think there should be more thought dedicated to communication. Yeah. Because… I think communication with the family is so important.’(F9)

‘…good internal communication, and positive feedback, positive reinforcement to staff, and I think clear indication about any changes to services and shortfalls, particularly around infection prevention and control, and any training on any new equipment that they use, or any changes to how treatment has to be delivered?’(S5)

‘You know, so I really think that, you know, should… it’s kind of looking at the bigger picture and how things could be opened back up?’(S6)

## 4. Discussion

This research was conducted to give residents, families, and staff at an RCS an opportunity to talk about their experience of restrictions during the COVID-19 lockdown period. The study also provides an opportunity to focus on learning for the future. The findings answered the research aim of this study, reflected in the themes related to the experiences of lockdown restrictions.

The four themes emerging from the data highlighted the challenges residents, families and staff faced due to lockdown restrictions. For residents, feeling safe was important and this was also a key concern for families, who relied on staff to protect their loved ones from contracting COVID-19. Staff were also deeply concerned with maintaining the safety of residents in their care, with some making significant personal sacrifices in order to avoid interaction outside of work, findings similarly reported in a Dutch study undertaken by Rutten et al. [13].

Older people bore the greatest burden of stress in terms of developing a serious illness or dying from contacting the virus. Ioannidis, Axfors, and Contopoulos-Ioannidis [14] state almost 96% of deaths due to COVID-19 infection worldwide occurred among those over the age of 70 years. Nevertheless, the residents in this study successfully coped with the challenges of the COVID-19 lockdown periods, despite the overwhelming fear portrayed in media reports of the human toll of COVID-19 outbreaks and deaths in RCSs in Ireland and internationally. Each of the residents had come to terms with the adversity and life challenges they individually experienced before their admission to the RCS. They expressed a sense of gratitude, safety and belonging, mainly because they had the company of staff throughout the experience. Additionally, the residents had confidence in the staff to care for them safely; accepting, respecting, and cooperating with the policy changes implemented by government and by the organisation as being benevolent, required and implemented in the resident’s best interests. This confidence may not have been the norm for other RSCs throughout the country. In a report by the Health Information and Quality Authority inspectors found some nursing homes did not have appropriate numbers of staff to care for residents [15]. Fifty eight percent of the nursing homes inspected were found to be non-complaint with governance and management regulations, stating these nursing homes were ill-equipped to manage the challenges presented by COVID-19. In our study, the residents’ experiences of dealing with their declining health and loss of independence, thereby requiring admission to the RCS, strengthened their wisdom and allowed them to transcend the difficulties and distress caused by the lockdown restrictions.

Given the limited understanding of COVID-19 when it first appeared there were genuine fear and concerns world-wide, meaning communication about the disease, its transmission and preventing its spread was of paramount importance. Communication with respect to the initial introduction of restrictions to visitors was generally handled well. However, concerns from family members arose with the level and focus of communications in respect of continuing restrictions and the slow reintroduction of visiting. In agreement with Sadler et al.’s [16] findings, strengthening communication supports when residential care facilities are locked down may reduce confusion and feelings of isolation for residents. Furthermore, Staker and Sun Choi [17] concur that families feel peace of mind when there are multiple communication channels for them to contact the resident. Yeh et al. [18] interviewed 156 family members of nursing home residents in Taiwan using a telephone survey and found most family members accepted and supported the visiting restrictions in place during the early waves of the global pandemic and family’s satisfaction with the overall quality of care their loved one received was independent of their acceptance of the visiting restriction policies. Similarly, we found there was genuine appreciation for staff from residents and their families who felt that staff had gone above and beyond what their job required. For families, staff were the link to their loved ones and for residents’ staff had become almost family, who were there to care for and assist them throughout this difficult and unprecedented period.

Isolation has adverse health consequences, negatively impacting mood, cognition, function, and quality of life [18,19,20]. In an effort to connect families with residents, window visits and virtual visits were implemented to mitigate the effects of loneliness and isolation from families and friends [21,22]. Window visits, where a person stands outside and speaks to the resident at safe distance through an open window [23] caused an amount of physical and emotional pain and was described by the residents as cruel, horrific, and awful, although they reconciled with the window visits as ‘being better than nothing’. In contrast to our findings, other researchers found window or outdoor visits to be tender moments giving residents ‘peace of mind’ [17] and for some, window visits was more appreciated for social interaction than virtual visits [24]. Most of the residents in the study worried more about their family members enduring the cold as they stood outside the resident’s windows, unable to touch or sometimes hear each other, more so than they worried about themselves. As restrictions lessened and the residents received the COVID-19 vaccines and booster doses, visiting restrictions eased based on risk assessments [23].

Hugelius, Harada and Marutani [24] report virtual visits using telecommunication apps were positively welcomed by family members, although there were mixed opinions about the use of technology from the residents in this study. The residents all had access and support to engage in virtual visits; however, they were not overly interested in virtual visits, some claiming the technology was too complicated. Hoffman et al. [25] and Giebel et al. [26] similarly found that older people sometimes found the mobile technology too complex to use. Rather, it was mostly families who communicated the need for virtual visits, reporting the challenges they experienced depending on the availability of staff or who was on duty that day. Sweeney et al., [27] reported similar findings, with family members placing greater emphasis on maintaining virtual contact with their relatives. Our findings reflect other qualitative research studies where relatives of older adults living in RCS reported feeling anxious and concerned about their family members’ psychological well-being during the lockdown period [18,27,28]. Staff also found the requests for virtual visits from visitors at times to be demanding and an extra burden on their workload, especially due to understaffing and competing caring priorities.

### Limitations

This qualitative study was conducted to develop an understanding of the experiences of residents, family members and staff in one RCS in Ireland. In line with qualitative research our findings cannot be generalized as the study took place in one site. There were no outbreaks of COVID-19 in the study site and no residents developed or died from COVID-19. The experiences of residents, staff and family members in other RCSs in Ireland and elsewhere where there were outbreaks and deaths from COVID-19 would no doubt yield different findings. A further limitation is the findings of this research study may only be applicable to the unprecedented lockdown restrictions as a consequence of COVID-19, prior to the availability of vaccinations. Additionally, for infection prevention and control reasons the time spent interviewing the residents was short. Moreover, there was a need to, based on ethical permission, exclude patients who did not have the capacity to give consent due to a diagnosis of dementia, and family members who were unable to communicate in English. The exclusion of these two groups could have an impact on learning more about social isolation of vulnerable groups during the COVID-19 restrictions.

Every effort was made to ensure trustworthiness in this research study. Lincoln and Guba’s [29] criteria of credibility, confirmability, transferability and dependability were used to establish trustworthiness.

## 5. Conclusions

The study highlighted the negative impacts of visiting restrictions on staff, residents and their family members during the COVID-19 lockdown in an Irish RCS. Infection prevention and control restrictions may be needed in residential care settings to manage infection outbreaks in the future; therefore, it is important to learn lessons from those who lived through and survived this experience.

Additional attention is needed if consideration of any reintroduction of restrictions occurs. A focus on maintaining a positive connection with family members using technology, with staff available to support residents to use the technology appropriately should also be considered. Enabling visits from residents’ family members using PPE must also be considered to overcome the difficulties experienced in respect of window visits.

Whilst staff working in RCS have learnt a significant amount about managing COVID-19, the pandemic is not as yet over and there is a continued requirement for vigilance to protect older residents and maintain their physical and psychological well-being. This research study was conducted during a time of heightened stress for residents, their family members and staff. There is scope for a further retrospective multi-site study, including RCS who had COVID-19 outbreaks, to develop a deeper understanding of the effects of a pandemic on the most vulnerable people in society. Older people, living in aged care facilities and those closest to them have a central role in identifying strategies that met their needs during the COVID-19 lockdown period, and their voices must be heard.

## Figures and Tables

**Table 1 ijerph-19-14002-t001:** Inclusion and exclusion criteria.

	*Inclusion Criteria*	*Exclusion Criteria*
** *Residents* **	Resident in the long-term care setting during COVID-19 lockdown restrictions (March–June 2020 and December 2020–May 2021). Has capacity to give informed consent. The Consultant Gerontologist agrees the resident is fit to participate, based on their clinical assessment. MMSE score of 21/30 and above, (MMSE score completed within six months prior to the study). Ability to communicate in English.	Residents with cognitive impairment who are unable to provide informed consent. Residents who were not living in the care facility during the COVID-19 lockdown. Residents who are unable to communicate in English.
** *Relatives* **	Family member of a resident living in the long-term care setting. Ability to give informed consent. Ability to communicate in English. Have the ability to engage in an online or telephone interview.	Unable to give informed consent. Unable to communicate in English.
** *Staff* **	Employed in the long-term care setting during the COVID-19 pandemic. Involved in provision of care to residents in the long-term care setting.	Staff who were not involved with the provision of direct resident care during the COVID-19 pandemic.

**Table 2 ijerph-19-14002-t002:** Demographics Details Residents.

*Resident (R)*	*Gender*	*Age*	*Length of Time Living in RCU*
*R1*	M	65–80	<1 year
*R2*	F	80+	1–3 years
*R3*	F	65–80	<1 year
*R4*	F	65–80	Over 5 years
*R5*	F	80+	1–3 years
*R6*	F	65–80	<1 year
*R7*	F	65–80	1–3 years
*R8*	F	65–80	1–3 years
*R9*	M	65–80	<1 year

**Table 3 ijerph-19-14002-t003:** Demographic Details Family Members.

*Family Member (F)*	*Age*	*Relationship to Resident*	*Length of Time Relative in RCS*	*Visits Pre-COVID-19 Restrictions*	*Visits during COVID-19 Restrictions*
*F1*	50–65	Sister	15 months	Daily	Visit 3 time a week, shared visits, 2 people max per visit; 1 h, but we do stay longer than an hour
*F2*	31–50	Wife	2 years	Every day, from 3 pm for 2 h	Visit 5 times a week because I emailed the Person in Charge. Initially 1 h, but I pushed for 2 h
*F3*	50–65	Husband	18 months	Family with their mother nearly 12 h a day	Visit 3 days per week, officially for 1 h but no pressure to leave
*F4*	50–65	Nephew and son	Aunt 2 years; Mother less than a year	Aunt had lots of friends, so didn’t need to visit too often. Mother came to RCU during COVID-19.	Window visits any time as granddaughter still can’t see grandmother and grand aunt. Window visits, sitting outside, worked as well as it could.
*F5*	65–80	Sister	less than a year	Anytime, no restrictions	Visit as needed
*F6*	31–50	Son	18 months	Big family always with mother. Someone visited every day at mealtimes and in the evening.	Window visits were variable depending on staff who were there at the time. No protocol for calls, may get a call, may not, depending on who was on duty.
*F7*	85+	Husband	9 months	Came to RCU during COVID-19 lockdown in 2020	Visit 4 times a week, lasting an hour
*F8*	65–80	Husband	less than a year	Came to RCU during the lockdown	Visit 4 days a week, one hour a day
*F9*	50–65	Daughter	18 months	1 h per day could visit as mother needed	Compassionate visits
*F10*	50–65	Son	18 months	1 h per day	Compassionate visits
*F11 + 12*	31–50	Daughter and Son	3 years	Very often with access 7 days a week between family, especially dad who visited daily	Visit 4 days a week for an hour between us all, 2 people allowed to visit 4 time a week, no children. Only one child can visit in the rose garden, and we need special permission for that

**Table 4 ijerph-19-14002-t004:** Demographic Details Staff.

*Staff (S)*	*Age*	*Years Working at RSC*	*Highest level of Education (Based on National Qualifications Framework)* *https://www.qqi.ie/what-we-do/the-qualifications-system/national-framework-of-qualifications (accessed on 29 September 2022)*
*S1*	31–49	6–10	Level 5
*S2*	50+	11–20	Level 7
*S3*	31–49	11–20	Level 9
*S4*	50+	20+	Level 9
*S5*	31–49	1 year	Level 9
*S6*	31–49	11–20	Level 9
*S7*	31–49	11–20	Level 5
*S8*	31–49	6–10	Level 5

**Table 5 ijerph-19-14002-t005:** Themes and sub-themes.

Themes	Sub-Themes
**‘Safety first’**	
**Communication**	‘The window visits’ Using technology helped. Feeling disconnected
**Staff going above and beyond**	
**What about the future?**	

## Data Availability

The data has not been made available through open source.

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
