# Peer review of "‘Safety First’: Residents, Families, and Healthcare Staff Experiences of COVID-19 Restrictions at an Irish Residential Care Centre"

_ijerph, 2022, doi:10.3390/ijerph192114002_

Round 1
Reviewer 1 Report
Thank you for the opportunity to review this article.
Introduction
Lines 33-34 cites the number of deaths. Until which date?
In line 54 there is a reference that appears within brackets and not as all references.
Lines 57-59 tackle inappropriate ageist comments, but it is unclear whether it is only on SNS or on other media as well. If it is only on SNS, than what about other media types?
Line 66 - TILDA. The term should be widely explained for those whom are not familiar with the TILDA study/database.
Method
Line 104 - all three references are not numbered correctly, it should be 10, 11, 12. Hence, all references should be re-checked.
the method section needs further explanation regarding the six-step method - why this method was chosen from all other qualitative analysis methods? In which components does this method is different from other qualitative thematic analysis methods?
Results
Overall, the results section lack a summary for each theme and sub-them.
Sub-paragraph 3.1 has different aspects of safety that appears in the citations and are not shown in the analysis. The authors need to analyze all the interviews (on this theme as well on other) more broadly. For example, in lines 139-142 the citation deals with safety but the main issue is not the safety but the interviewee being a prisoner.
The same with the citation in lines 150-152, it is a different citation that deals with safety as a "must have".
All the citations in the results' section are too short, and it is comprises the ability to understand the context. Qualitative study seek to bring the interviewees' "story" or "voice" and here it is not happening.
Sub-theme 3.2.2 - the theme lacks context or explanation. The tenants doesn't have mobile phones? why? what is the situation in Ireland regarding mobile phone usage?
Citations in lines 232-235 talks about companionship more than communication.
Theme 3.2.3, again, there are different levels of disconnectedness within citations and the authors doesn't tackle it.
The citation in lines 312-314 is different from other citations in the theme.
What is explained in lines 316-318 is not demonstrated within citations.
Overall, the results' section is not thorough.
Discussion
Somehow, almost all of the issues that are mentioned in the discussion, are not shown, or shown very shortly, above surface, within results' section.
Also, around lines 385-392: what do the literature says about what the authors mention?
What appears in lines 396-404 does not reflected within citations.
Conclusions
Lines 431-432, the study does not highlight only what the authors mention, there are some other important issues that emerge from it.
Author Response
The authors wish to thank the reviewer for their comments.
The response to each comment is provided in the attached.

Reviewer 2 Report
The topic is very interesting and it was one of the main concern and problem about the management of COVID infection among elderly people.
But the sample chosen was to small and the analisys and presentation of data may be improved and presented in a more synthetic way.
Author Response

(The authors gave the same response as above.)

Reviewer 3 Report
Minor comments:
-Line 53: Please change resident's to residents' since the plural form needs to be used here.
Major comments:
-Table 1: I understand why patients with dementia and family members who are unable to communicate in English were excluded from this study that was based on qualitative interviews, but these 2 groups were likely among the most socially isolated and the most likely to experience morbidity and mortality due to COVID. I would recommend adding recognition of this to the limitations section.
-Section 2.3: The authors mention that they conducted virtual visits. Would this have potentially selected for participants of higher socioeconomic status with access to the Internet and a computer, thus making it less likely that voices from patients with lower SES would have been heard in the qualitative data?
-Section 3.3: The high level of dedication of staff members may make this facility somewhat different than other facilities that suffered from staffing shortages and low morale during the COVID pandemic.
-Discussion (paragraph 3): The high confidence that patients placed in staff might also not be representative of most nursing facilities. The authors might wish to comment on this.
-Discussion (paragraph 5): The description by some respondents of the window visits as "horrific" seems different than media portrayals of window visits that seemed to portray these visits as tender moments to connect with family.
-Discussion (paragraph 6): I think one of the most important findings of this paper is that the residents of this nursing facility were not particularly fond of virtual communication and that they sensed it was more of a benefit for the family members than for themselves. I also think the finding that staff found that virtual visits exacerbated their workload is an important finding. Both findings could be applicable to how to structure communication when patients need to be isolated and how this communication should occur during the next pandemic.
Limitations: Again, this nursing home seems like an outlier since there were no cases of COVID detected at this site, which further limits generalizability of the findings. This reviewer recognizes that this is out of the authors' control.
Conclusion: I would like the authors to comment on future directions that they envision, if any, with this research such a potentially conducting retrospective qualitative interviews among a larger number of nursing homes, including those who had COVID outbreaks. It may also be interesting to compare qualitative findings between nursing homes that had COVID outbreaks and those that did not.
Author Response
The authors thank the reviewer for their comments.
The attached contains a response to each comment.

Round 2
Reviewer 2 Report
The revisions had improved the quality of the paper that may be published